# Scribbles for All: Benchmarking Scribble Supervised Segmentation Across Datasets

**Wolfgang Boettcher**[1]   **Lukas Hoyer**[2]   **Ozan Unal**[2]   **Jan Eric Lenssen**[1]   **Bernt Schiele**[1]

[1] Max Planck Institute for Informatics, Saarland Informatics Campus, Germany
[2] ETH Zürich, Switzerland
{wolfgang.boettcher, jlenssen, schiele}@mpi-inf.mpg.de
{lhoyer, ouenal}@vision.ee.ethz.ch

## Abstract

In this work, we introduce *Scribbles for All*, a label and training data generation algorithm for semantic segmentation trained on scribble labels. Training or fine-tuning semantic segmentation models with weak supervision has become an important topic recently and was subject to significant advances in model quality. In this setting, scribbles are a promising label type to achieve high quality segmentation results while requiring a much lower annotation effort than usual pixel-wise dense semantic segmentation annotations. The main limitation of scribbles as source for weak supervision is the lack of challenging datasets for scribble segmentation, which hinders the development of novel methods and conclusive evaluations. To overcome this limitation, *Scribbles for All* provides scribble labels for several popular segmentation datasets and provides an algorithm to automatically generate scribble labels for any dataset with dense annotations, paving the way for new insights and model advancements in the field of weakly supervised segmentation. In addition to providing datasets and algorithm, we evaluate state-of-the-art segmentation models on our datasets and show that models trained with our synthetic labels perform competitively with respect to models trained on manual labels. Thus, our datasets enable state-of-the-art research into methods for scribble-labeled semantic segmentation. The datasets, scribble generation algorithm, and baselines are publicly available at https://github.com/wbkit/Scribbles4All.

## 1   Introduction

Semantic segmentation is one of the most crucial tasks for computer vision research and a key component to scene understanding in many applications. While substantial advancements have been made in designing highly accurate segmentation architectures [25, 6, 8, 7, 17], those models heavily rely on detailed labels. The crafting of such dense labels constitutes a laborious and resource-intensive process. This limitation impedes the availability of specialized datasets and the practical deployment of vision algorithms in real-world scenarios. It is particularly pronounced in self-driving applications, which demand immense amounts of training data [2, 32, 9] or in domains with fine-grained classes [24, 54]. Those settings necessitate models capable of dealing with complex inter-object relations, varying shapes, and scales. Even in the current era of large pre-trained foundation models [27, 19], fine-tuning these models on custom, labeled data still remains a necessity for many applications [48, 4, 42].

One popular approach to addressing the issue of label cost is weakly supervised semantic segmentation (WSSS) [3, 49, 43] which has made significant progress in recent years. WSSS methods use incomplete labelling in the form of image level labels [51] or bounding boxes [18, 26, 50] to label

38th Conference on Neural Information Processing Systems (NeurIPS 2024) Track on Datasets and Benchmarks.

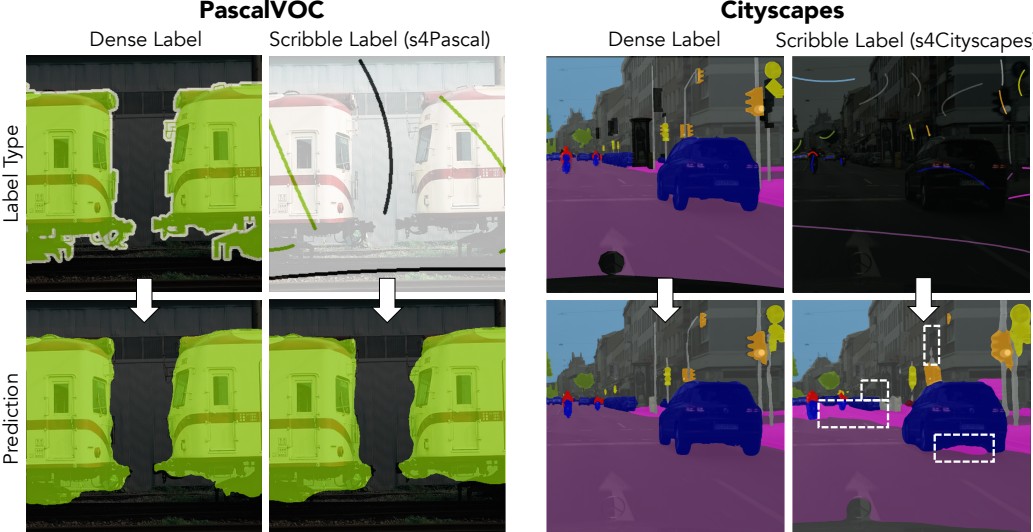

Figure 1: **Visual difference in scribble-supervised performance** – While predictions from scribble supervised models are almost identical to fully supervised models for PascalVOC, the quality of segmentation for scribble supervised Cityscapes models is visibly poorer (see dotted boxes), highlighting the greater complexity of the dataset and the need for further research.

objects. Similarly, Sparsely Annotated Semantic Segmentation (SASS) is defined by labelling a subset of the image pixels through coarse labels [11, 10], labelling every object through points [1] or drawing scribbles [23, 37]. Previous research has shown that especially scribbles are a promising means of attaining cheap yet powerful labels [23, 40]. While they take only slightly longer for the annotator to label than points, they have been shown to enable stronger segmentation results [45, 21]. Annotating full segmentation images takes several minutes for object centric datasets like Pascal [24] or even hours for more complex driving scenes [9, 30] while labelling ScribbleSup [23] took on average about 25 seconds. Moreover, state-of-the-art methods gain 7–10% mIoU on PascalVOC [14] for scribble labels over point labels [31, 45, 21]. This success has lead to a series of promising training methods for SASS with scribbles [13, 39, 44].

However, currently, there exists only one popular segmentation dataset with scribble labels, namely ScribbleSup, introduced by Lin et al. [23] for the PascalVOC dataset. An example image of it is shown in Fig. 2. Two challenges arise for the research area of scribble-supervised segmentation methods. Firstly, generalization of methods to other datasets cannot be verified. Secondly, PascalVOC is too easy to serve as the sole benchmark for scribble-supervised methods as visualized in Fig. 1. It consists mostly of images with one object class and the background class. By learning precise class boundaries of the dominant background class, a model can already achieve high performance while the challenge of learning object-to-object boundaries is less relevant. In contrast, modern semantic segmentation faces additional challenges such as small object instances (e.g. poles in Cityscapes) or a large number of semantic classes (e.g. 150 classes in ADE20K), which cannot be properly benchmarked with PascalVOC (see Fig. 1).

We present an algorithm that derives scribble labels from fully labeled datasets which are functionally equivalent to hand-drawn scribbles allowing us to bring scribble-supervision to a variety of popular and challenging segmentation datasets. Our datasets enable future research on segmentation methods trained or fine-tuned on scribble labels. Explicitly, we extend the applicability of scribble-based segmentation methods to the broad range of available datasets such as Cityscapes [9], ADE20K [54], KITTI360 [22], and more. Our main contributions can be summarised as follows:

- We present an automatic scribble generator for any fully labeled segmentation dataset, enabling future research into state-of-the-art models for scribble-supervised segmentation.
- We introduce s4Pascal, s4KITTI360, s4Cityscapes and s4ADE20K, four automatically generated datasets with scribble labels.
- We benchmark state-of-the-art segmentation methods on our datasets, showing that models trained with our scribbles perform on-par with models trained on manually created scribbles.

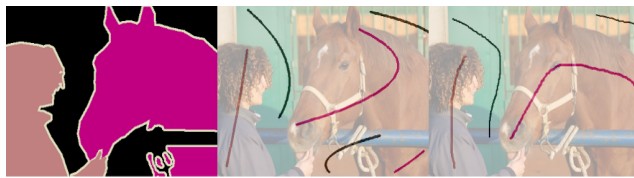

Figure 2: **Overview of label types** – Left to right: Full PascalVOC semantic label, scribble labels created by our scribble generation algorithm for s4Pascal, hand-drawn scribble labels from ScribbleSup.

## 2 Related Work

In this section, we provide a structured overview of the research that is of particular relevance for our work and the current state-of-the-art with respect to scribble labelled datasets.

**Weakly Annotated Semantic Segmentation (WASS)** Methods for WASS have been trained using image-level labels [29, 49, 43] or bounding-box labels [18, 26]. While image-level labels are fast to obtain, they suffer from the lack of pixel-level information, which renders them unsuitable for complex scenes. Bounding-boxes offer spatial supervision on individual objects but fail to deal with overlapping, not box-aligned objects. In contrast to WASS labels, scribbles provide a better supervision signal and are cheaper to obtain than bounding boxes [45]. The latter suffer from added challenges such as the overlap of boxes.

**Sparsely Annotated Semantic Segmentation (SASS)** Related to supervision with weak labels, SASS is concerned with using labeled pixel subsets, allowing direct supervision on sparse regions of the image. The two main label types for SASS are point labels [1] and scribbles [23], which led to several follow-up representations [11]. Depending on the requirements and domain, those other labelling strategies are also used, such as coarse annotation [11, 10]. The latter comes with the benefit of being more expressive but also requires more effort. Scribbles have been demonstrated to be a Pareto-optimal choice between labelling effort and segmentation quality. Unal et al. [36] introduce ScribbleKITTI for SemanticKITTI and demonstrate that scribbles can lead to approximately supervised performance in the 3D domain. Scribbles, traditionally, require a human annotator, which entails the need for further resources to provide the respective labels, hindering model research and development. In this work, we propose a method to automatically generate scribble labels from dense 2D annotations, providing excellent benefits for the development of future SASS methods.

**Training SASS models.** In contrast to a dense semantic label, the labels used for SASS do not provide information on the object outline, leading to challenges in class boundary estimation [23]. Several methods utilise auxiliary tasks such as edge detection [40] or saliency [52] to improve performance. However, those methods tend to suffer from model errors in the auxiliary task, which limit their prediction capabilities [21]. Other approaches [34, 33, 37] use regularised losses that model interdependencies of the labeled and unlabeled pixels. Often, these are combined with CRFs [20], which are adapted for region growing on the pseudo labels generated by the model and for overall refinement of the predictions [49, 41, 44]. For example, URSS [28] addresses the inherent model uncertainty in semi-supervised training by applying random walks and consistency losses in a self-training framework. Similarly, Valvano et al. [38] uses self-training but with multi-scale consistency while TEL [21] achieve performance increases by introducing a similarity prior through a novel tree-based loss. Most recently, SASformer [31] utilises the attention mechanisms in transformer architectures, using global dependencies to achieve more accurate segmentation results, marking the trend away from auxiliary tasks and two-stage approaches towards end-to-end trainable frameworks. To validate our automatic scribble labels, we train and evaluate a subset of these recent methods on our datasets and show performance comparative to manual labelling.

**Scribble Annotation Datasets** We present the second scribble image dataset so far, only preceded by ScribbleSup [23]. ScribbleSup only contains labels for the older PascalVOC [14] dataset. In contrast, we provide labels for a larger set of currently relevant datasets and, additionally, a method for the automatic generation of scribbles for any fully-annotated dataset.

## 3 Automated Scribble Generation

This section introduces our automatic scribble generation method, before the individual generated datasets are described in Sec. 4. We begin by shortly outlining design objectives in Sec. 3.1, before describing the detailed algorithm in Sec. 3.2.

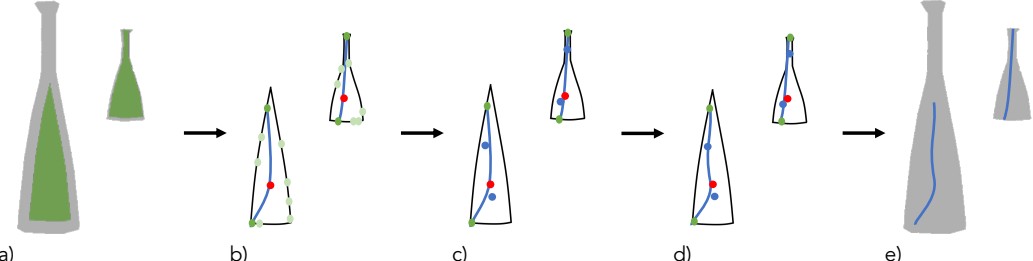

Figure 3: **Scribble Generation** – a) Size dependent erosion, b) COM in red, sampling of points on the edge in green, determination of the approx. farthest pair in darker green and tentative scribble in blue c) Sampling of two extra points along the tentative scribble d) Fitting final scribble through points e) Scribble overlayed on initial segmentation map.

## 3.1 Design Objectives

The presented scribble generation algorithm takes an image with corresponding dense segmentation labels as input and produces a single scribble, represented as a set of points, for each object in the image. We formulate the following design objectives:

1. **Mimic human annotations.** The generated scribbles should approximately resemble scribbles that human annotators draw. Specifically, they are supposed to be more coarse for larger, simple geometries and more precise for detailed objects, as would be the case with hand-crafted labels. Also, the scribble is expected to go roughly through the centre part of an object and not to come too close to its margins for a large portion of its length.
2. **Probabilistic generation.** The generation of labels should occur in a probabilistic fashion to prevent mean collapse when confronted with similar shapes, maintaining enough variance in the labelling process.
3. **No boundary violation.** We also apply hard constraints that prevent scribbles from violating any class boundaries.

We design the algorithm described in Sec. 3.2 to reach these objectives.

## 3.2 Scribble Generation Algorithm

This section describes the scribble generation algorithm, as detailed in Alg. 1 and visualized in Fig. 3.

**Preprocessing.** The algorithm commences by separating the semantic mask $GT_{mask}$ of an image by class $C$. Then, we apply class-wise connected component analysis to obtain separate masks for each instance of the respective class. Each object mask $b \in B$ of $J$ total objects masks is subsequently subjected to morphological erosion in an amount $\epsilon_1$ depending on the object area to comply with design objective 1 as shown in Fig. 3a). If an object is separated into multiple masks at this stage, each mask will be considered an individual object for further processing. Through this, we ensure that objects with complex non-convex shapes are properly labeled as well by generating multiple scribbles. If two objects overlap in a single connected component, they are considered as a single instance. This splitting procedure is only applied in the first erosion step. After the following ones (see below), smaller separated objects are removed instead.

**Polynomial fitting.** After the preprocessing procedure, the algorithm fits a curve through each obtained mask by iteratively repeating the following process until a valid scribble annotation is found or the blob is completely eroded: A fixed-rate $\epsilon_2$ binary erosion is applied. After that, the edge image of the blob $e$ and its centre-of-mass (COM) $c_M$ is calculated. In the case of strongly non-convex shapes, it can occur that the COM is not part of the image. If that happens, the object is subjected to a skeletonisation operation in the style introduced by Zhang et al. [53] and the closest point in terms of $L_2$-distance of the skeleton wrt. to the true COM is used as a COM substitute.

Hand-drawn scribbles usually follow the object's predominant direction up to some human noise factor. To imitate this behaviour, we randomly sample $Q$ points on the object edges and select a pair $(p_1, p_2)$ with the maximum distance $\|p_1 - p_2\|_2$ by farthest point sampling. The pair spans the furthest distance of the object most of the time while retaining a chance of suboptimal solutions and

**Algorithm 1** Scribble Generation

**Input:** $GT_{mask} \in \mathbb{N}_0^{m \times n}$

$\quad C_{mask} \leftarrow \text{separateClass}(GT_{mask})$ $\hfill \triangleright\, C_{mask} \in \mathbb{Z}_2^{C \times m \times n}$

$\quad B \leftarrow \text{componentAnalysis}(C_{mask})$ $\hfill \triangleright\, B \in \mathbb{Z}_2^{J \times m \times n}$

$\quad \textbf{for } b \in B \textbf{ do}$ $\hfill \triangleright\, b \in \mathbb{Z}_2^{m \times n}$

$\qquad \hat{b} \leftarrow \text{binaryErosion}(b, \epsilon_1(\text{area}(b)))$ $\hfill \triangleright\, \hat{b} \in \mathbb{Z}_2^{m \times n}$

$\qquad \textbf{while } \hat{b} \neq \emptyset \textbf{ do}$

$\qquad\quad \hat{b} \leftarrow \text{binaryErosion}(\hat{b}, \epsilon_2)$

$\qquad\quad e \leftarrow \sqrt{(S_x * \hat{b})^2 + (S_y * \hat{b})^2}$ $\hfill \triangleright\, S \text{ - Sobel operator, } e \in \mathbb{Z}_2^{m \times n}$

$\qquad\quad c_M \leftarrow \text{centerOfMass}(\hat{b})$ $\hfill \triangleright\, c_M \in \mathbb{Z}_+^2$

$\qquad\quad \textbf{if } c_M \notin \hat{b} \textbf{ then}$

$\qquad\qquad c_M \leftarrow \text{skeletonCOM}(\hat{b}, c_M)$ $\hfill \triangleright \text{ Closest pt. in } L_2\text{-dist. to COM in skeleton of } \hat{b}$

$\qquad\quad \textbf{end if}$

$\qquad\quad \textbf{for } i \in N \textbf{ do}$

$\qquad\qquad P \leftarrow \text{sample}(e, Q)$ $\hfill \triangleright\, Q \in \mathbb{Z}_+, P \in \mathbb{Z}_+^{Q \times 2}$

$\qquad\qquad (p_1, p_2) \leftarrow \text{furthest-pair}(P, L_2)$ $\hfill \triangleright \text{ max. dist. by } L_2\text{-norm, } p_1, p_2 \in \mathbb{Z}_+^2$

$\qquad\qquad \hat{c}_M \leftarrow c_M + \epsilon, \quad \epsilon \sim \mathcal{N}(0, \sigma_{com})$

$\qquad\qquad \hat{\beta}_0, \hat{\beta}_1, \hat{\beta}_2 \leftarrow \arg\min_{\beta_0, \beta_1, \beta_2} || \sum_{j \in \{p_1, p_2, \hat{c}_M\}} j_y - (\beta_0 + \beta_1 j_x + \beta_2 j_x^2) ||^2$

$\qquad\qquad F_2(x) = \hat{\beta}_0 + \hat{\beta}_1 x + \hat{\beta}_2 x^2$

$\qquad\qquad \sigma \leftarrow \text{area}(\hat{b})/\alpha$

$\qquad\qquad (p_3, p_4) \leftarrow \text{sample}(F_2(x), 2) + \epsilon, \quad \epsilon \sim \mathcal{N}(0, \sigma)$ $\hfill \triangleright \text{ Sample from polynomial } F_2$

$\qquad\qquad \hat{\beta}_0, \hat{\beta}_1, ..., \hat{\beta}_4 \leftarrow \arg\min_{\beta_0, ..., \beta_4} || \sum_{j \in \{p_1, ..., 4, \hat{c}_M\}} j_y - (\beta_0 + ... + \beta_4 j_x^4) ||^2$

$\qquad\qquad F_4(x) = \hat{\beta}_0 + \hat{\beta}_1 x + ... + \hat{\beta}_4 x^4$

$\qquad\qquad \textbf{if } F_4(x), x \in [p_1, p_2] \subset \hat{b} \textbf{ then}$

$\qquad\qquad\quad \textbf{break}$

$\qquad\qquad \textbf{end if}$

$\qquad\quad \textbf{end for}$

$\qquad \textbf{end while}$

$\quad \textbf{end for}$

$\quad \textbf{return } F_4(x), x \in [p_1, p_2]$

variance like in the case of a human annotator. Random sampling allows the algorithm to explore an extended solution space if the scribble generation takes multiple iterations for the object. The returned point pairs and the COM with added noise $\hat{c}_M$ are utilised to obtain a second-order polynomial $F_2$ as depicted in Fig. 3b), by solving the linear least-squares problem. We choose $x$ as the coordinate with the largest distance between $p_1$ and $p_2$.

The next step improves the label variance of the scribble shapes and satisfies design objective 2. For this, two points $p_3, p_4$ are sampled from $F_2$, adding area-dependent Gaussian noise. As the final step, $\hat{c}_M, p_1, ..., p_4$ are used to find a 4th-order polynomial $F_4$ as shown in Fig. 3d).

If the curve is entirely within the blob and does not violate any class boundaries (objective 3), the label is considered valid. Otherwise, the process of sampling from the edge and fitting curves is repeated up to $N$-times. If this does not yield a valid scribble, the algorithm is restarted.

# 4 Automatic Scribble Datasets

We now describe the datasets we created with the algorithm presented in Sec. 3.2. All datasets are publicly available and can be used to advance research in the area of scribble-supervised models.

**s4Pascal Dataset** The Pascal VOC 2012 dataset [14] is a widely used benchmark dataset in the field of object detection, image classification and semantic segmentation. It consists of images collected from various sources and covers 20 object classes and the background class. In terms of size, the Pascal VOC 2012 dataset for semantic segmentation contains 10,582 training images and 1,449 validation images when using the augmented version introduced by Hariharan et al. [15]. The dataset's semantic mask includes a "do not care" label that is applied between class boundaries and fine-grained structures of the same class.

Table 1: **Quantitative Evaluation of the Scribbles for All (s4) datasets** – When compared, ScribbleSup and s4Pascal show very similar characteristics with respect to the number of labeled pixels and the number of labeled pixels that are within a defined distance to the class boundary. The average number of scribbles per image differs, however.

| Dataset | | | | | Statistical Property | | | |
|---------|------|-----------|--------|-------------|----------------|----------------|----------------|----------------|
| | ours | train/val | # cls. | % px. lab. | $\alpha_{10px}$ | $\alpha_{20px}$ | $\alpha_{30px}$ | avg. scribbles |
| ScribbleSup | | 10,582/1449 | 21 | 2.07 % | 11.3 % | 27.49 % | 44.38 % | 3.89 |
| s4Pascal | ✓ | 10,582/1449 | 21 | 2.25 % | 10.1 % | 30.08 % | 46.50 % | 5.11 |
| s4Cityscapes | ✓ | 2975/500 | 19 | 2.36 % | 13.8 % | 27.45 % | 35.24 % | 42.27 |
| s4KITTI360 | ✓ | 49,000/12,000 | 16 | 2.49 % | 5.09 % | 16.93 % | 29.80 % | 14.94 |
| s4ADE20K | ✓ | 25,574/2,000 | 150 | 4.71 % | 4.35 % | 9.10 % | 12.03 % | 17.27 |

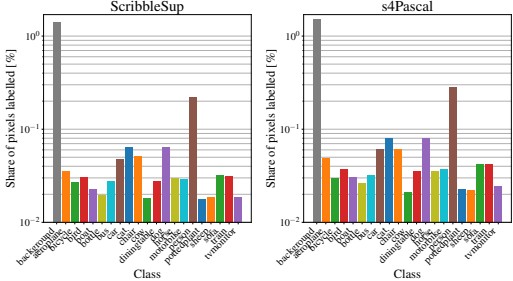

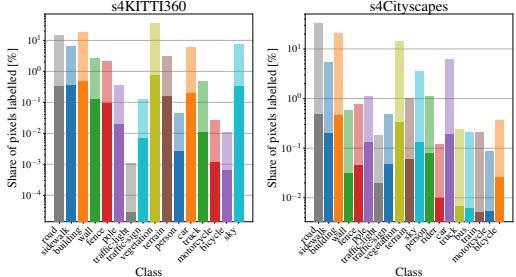

Figure 4: **Class distribution for s4Pascal and ScribbleSup** – The object class distribution wrt. to the entirety of GT-pixels is very similar with the exception of the background class that is more heavily labeled in the s4Pascal dataset.

Figure 5: **Class distribution for s4KITTI360 and s4Cityscapes** – The share of pixels labeled for each class is depicted in bright colors while the transparent bars represent the share of that class in the fully supervised dataset.

The scribble labels for PascalVOC were introduced by Lin et al. [23] as the *ScribbleSup* dataset. Fig. 2 depicts a segmentation map and scribble labels from ScribbleSup on the right. Given that ScribbleSup is the only available dataset with hand-crafted scribble labels and provides dense semantic maps as well, it is the suitable reference to evaluate the synthetic scribble generation algorithm described in Sec. 3. Dense segmentation maps are required since those serve as the input for the synthetic scribble generation. The new synthetic scribble dataset is created using the dense labels of the Pascal VOC 21-class dataset. It is from now on referred to as *s4Pascal*. Label generation includes all object classes as well as the background class. The "do not care" areas of the segmentation maps are omitted. Therefore, those are also not valid points for the scribble generation algorithms, while the hand-crafted labels appear to have no clear policy on whether scribbles are allowed to intersect "do not care" regions. Overall, s4Pascal adds synthetic scribbles to the PascalVOC dataset, allowing for model training with three different types of segmentation labels.

In general, the scribble labels of ScribbleSup and s4Pascal are statistically very similar. As shown in Tab. 1 both contain approximately the same share of labeled pixels and exhibit similar closeness of scribbles very close to the class boundaries ($\alpha_{10px}$). The same is true for the overall spatial distribution of scribbles within the objects the label ($\alpha_{20px}$, $\alpha_{30px}$) corresponds to. Being this similar, their visual appearance is close as well, as depicted in Fig. 2. Likewise, the class-wise distribution of scribbles depicted in Fig. 4 shows no significant aberrations when comparing the two label sets. Consequently, the scribbles generated by our scribble algorithm mimic the human-annotated labels closely. The only significant difference lies in the average number of scribbles used to label each image as shown in Tab. 1. This behaviour is explained by the human scribbles partially drawing over the don't care areas of the dataset which is prohibited for the algorithm and for more complex objects, the automatically generated labels may be broken down into multiple scribbles while the human may draw a single more complex line.

Importantly, as can be seen from Tab. 2 and as discussed in Sec. 5.2, the results obtained with ScribbleSup and s4Pascal are similar for various segmentation algorithms, further highlighting the similarity of our generated scribbles to human scribbles.

Table 2: **Quantitative comparison of SOTA methods on s4-datasets** – Alongside the absolute performance expresssed in mIoU, we also list the segmentation results with respect to the fully supervised model (Sup.). The methods are grouped by their respective encoder backbone. The * marks values taken from literature.

| Dataset | | | mIoU ↑ | | | | |
| --- | --- | --- | --- | --- | --- | --- | --- |
| | | | *SegFormer-B4 [46]* | | *DeepLabV3+ [5]* | | |
| | Ours | *Sup.* | EMA | SASformer [31] | *Sup.* | TEL [21] | AGMM [45] |
| ScribbleSup [23] | | 86.6 | 79.2 | **79.5*** | 84.6* | 76.4 | 76.8 |
| s4PASCAL | ✓ | 86.6 | 78.8 | **79.0** | 84.6* | 76.8 | 73.8 |
| s4Cityscapes | ✓ | 83.8* | **67.7** | 65.8 | 78.3* | 64.5 | 56.6 |
| s4KITTI360 | ✓ | 66.6 | 57.4 | 49.7 | 64.8* | **59.7** | 49.6 |
| s4ADE20K | ✓ | 51.1* | 37.7 | **41.4** | 46.8* | 39.6 | 35.4 |
| | | | **Rel. Performance ↑** | | | | |
| | Ours | *Sup.* | EMA | SASformer | *Sup.* | TEL | AGMM |
| ScribbleSup | | - | 91.5 % | 91.8 % | - | 90.3 % | 90.7 % |
| s4PASCAL | ✓ | - | 91.0 % | 91.2 % | - | 90.8 % | 87.2 % |
| s4Cityscapes | ✓ | - | 80.8 % | 78.5 % | - | 82.4 % | 72.2 % |
| s4KITTI360 | ✓ | - | 86.2 % | 74.6 % | - | 92.1 % | 83.1 % |
| as4ADE20K | ✓ | - | 73.8 % | 80.4 % | - | 84.6 % | 69.3 % |

**Further Automatic Scribble Labeled Datasets** The lack of different scribble labeled datasets impedes thorough benchmarking of scribble-supervised methods under different domains. To alleviate this, we apply the scribble generation algorithm to a selection of popular segmentation datasets. Labeling these provides the foundation for benchmarking SOTA methods in the next step. We introduce *s4Cityscapes* which is a set of scribble labels for the Cityscapes dataset [9] that is known for the broad range of object scales and object sizes it requires the segmentation model to learn. Due to the high level of detail in the data, the scribble algorithm is parameterized such that also small objects are labeled. Furthermore, we provide *s4KITTI360* which contains scribble labels for KITTI360 [22] which like Cityscapes is a self-driving domain dataset. In contrast to the latter, it contains a notably higher number of labeled images but a lower level of detail in ita annotations. Like Pascal, those datasets only contain a small number of object classes. It is also important to asses how models can cope with fine-grained classes. Thus, we further provide scribble labels *s4ADE20K* for the ADE20K dataset [54], which consists of 150 classes.

The different properties of these datasets translate into different dataset statistics for the automated scribbles. While the number of scribbles per scene is similar for ADE20K and KITTI360, the share of labeled pixels is higher for ADE20K due to the lower image size as shown in Tab. 1. The dominance of furniture-related classes, doors, windows and other convex geometric objects further leads to s4ADE20K having scribbles with a relatively high distance to class boundaries. When looking at the two self-driving datasets, the higher level of detail and small objects in Cityscapes becomes apparent as the average number of scribbles per image is more than double that of KITTI360. The higher prevalence of slim objects such as poles and traffic-lights/signs moreover entails greater closeness to class boundaries. Further details on the class distribution are visualised in Fig. 5.

The datasets generated in this work are chosen to demonstrate the versatility and usefulness of the proposed scribble generator. The algorithm can be applied to all pre-existing datasets that contain dense segmentation maps, making it universally applicable. Further information on the algorithm runtime per dataset conversion is included in App. A.3.

## 5    Experiments

In this section, we perform evaluations on the proposed datasets. The section begins with providing implementation details in Sec. 5.1, before presenting segmentation experiments in Sec. 5.2, a scribble length ablation in Sec. 5.3, and a discussion about limitations in Sec. 5.4.

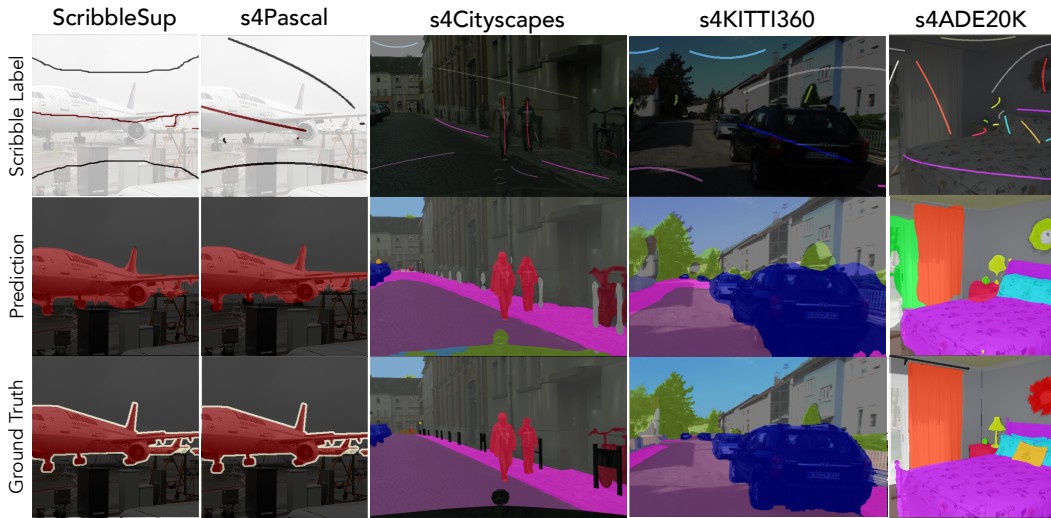

Figure 6: **Qualitative performance on the s4-datasets and ScribbleSup** – Shown are the input image overlaid with the corresponding scribbles, the EMA-model prediction and the ground truth. Color legends can be found in App. A.1.

## 5.1 Implementation Details

We evaluate the s4-scribble datasets using three current SOTA methods, namely Tree Energy Loss (TEL) [21], AGMM-SASS [45] and SASformer [31]. Both ScribbleSup and s4Pascal are trained according to the information provided by the authors and the published code. Hyperparameters for s4ADE20K and s4Cityscapes were also kept at the values described in TEL and AGMM-SASS. For training ADE20K and Cityscapes on SASformer, we used the same learning rates as in TEL. For KITTI360 we found the hyperparameters for Cityscapes to be permissive. All models were trained on four RTX 8000 GPUs. All models were trained in a single-stage process without postprocessing such as applying CRFs. More information is provided in App. A.5.

Additionally, we also train on a simple mean-teacher [51, 47, 35] setup with a Segformer-B4 [46] backbone to provide a naive WSSS baseline that does not make any specific prior assumptions like specialised losses, TEL, or architecture dependencies, SASformer. The training procedure was kept as published for SegFormer with additional augmentations for the student through CutOut [12] and AugMix [16]. The loss is composed of an equally weighted supervised loss from the scribbles and a KL-divergence loss informed by the teacher.

## 5.2 Baseline Scribble Datasets

As revealed by Tab. 2, the majority of three of the four methods used for comparing the ScribbleSup and s4Pascal datasets leads to very similar segmentation results differing 0.5 % mIoU or less. While EMA, and SAS perform better on ScribbleSup, TEL performs better on s4Pascal. The only method where the two label sets lead to different results is AGMM-SASS which shows a decline of approx. 3 % when trained with s4Pascal showing that some SASS methods are more susceptible to changes in label distribution than others. In conclusion, these results validate our observation from Sec. 4 that our scribble generation algorithms produce labels that are almost equivalent to human-created scribble labels. For class-wise evaluation, refer to App. A.4. Comparing scribble-supervised with fully-supervised models, the two methods with a SegFormer-B4 backbone show about 92 % relative performance, while the methods with ResNet101/DeepLabv3+ architectures both reach about 90 % relative performance as listed in Tab. 2. There is no relevant performance difference on ScribbleSup between the different methods that share the same backbone architecture, giving further evidence that the current benchmark is saturated.

This insight is supported by the results obtained from experiments on the s4ADE20K, s4Cityscapes and s4KITTI360 datasets. The typical relative performance drops to approx. 80 % as shown in Tab. 2. The more challenging datasets also lead to reduced mIoU values in absolute terms which is perceivable

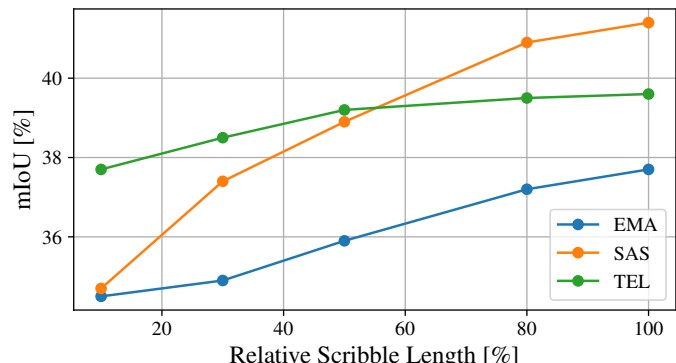

Figure 7: **Effect of scribble length on prediction performance** – The different methods exhibit varying robustness with respect to changes in the scribble length, evaluated here for the s4ADE20K dataset. The TEL method is less strongly affected than the naive EMA-model, while the effects on SASformer are the most severe. This highlights the importance of scribble length ablations for comprehensive method benchmarking.

on the examples shown in Fig. 6. Additionally, clear disparities between method performances can be observed as well as different best relative performances depending on the dataset. For instance, the high number of training images enables TEL to reach more than 90 % relative performance on s4KITTI360 while the best value for the similar but smaller dataset s4Cityscapes is approx. 10 % mIoU lower. We therefore hope that our datasets can facilitate research into scribble-labeled semantic segmentation models towards closing this performance gap.

### 5.3  Scribble Length Ablations

Shrinking the initial scribbles to different proportions leads to varying deterioration of prediction results depending on the applied methods. Fig. 7 illustrates that TEL is the most robust towards reductions in scribble length dropping by less than 2 % mIoU as the scribble length is reduced to a tenth of the original size. More sensitive is SASformer with a decline of about 7 % mIoU. While not being the best-performing SOTA method for full scribble lengths and small reductions, TEL leads to better results for stronger degradations. The naive mean teacher shows similar behaviour to SASformer though less pronounced. These results illustrate the importance of scribble length ablations when benchmarking methods to aid in obtaining a more thorough understanding of how the developed methods react to label variations.

### 5.4  Limitations

The proposed automatic scribble generation algorithm requires a dataset with existing semantic segmentation ground truth. As the purpose of this paper is a more broad evaluation of scribble-supervised methods on common segmentation scenarios, feasible basis datasets are available for many domains. However, for new or custom use cases, this assumption might not hold and manual scribble annotations can be necessary.

## 6  Conclusion

We presented new scribble annotations for four popular semantic segmentation datasets and a generally applicable method to generate those. Our work has shown that using more complex datasets reveals a widening gap between full supervision and scribble SOTA methods, compared to previous datasets. Furthermore, we demonstrated that different methods cope differently well with challenges such as shorter scribbles. Therefore, we suggest expanding the evaluation procedures for scribble-supervised segmentation to multiple datasets and also providing scribble-length ablations to show the robustness of the methods. We hope that our s4-datasets will drive a robust benchmarking of future scribble-supervised methods to close the gap to densely-supervised segmentation training. In particular, we see a strong potential in adapting vision foundation models with scribbles to custom applications.

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

# A  Supplementary Material

## A.1  Color Legends

This section provides the color legends for the segmentation maps and ground-truth images that are displayed in Fig. 6 of the main paper. Colormaps are provided for (s4)ADE20K, (s4)Cityscapes and (s4)KITTI360. As only two classes, background and areoplane, are present in the (as)PascalVOC example, no additonal colort legend is provided for this dataset.

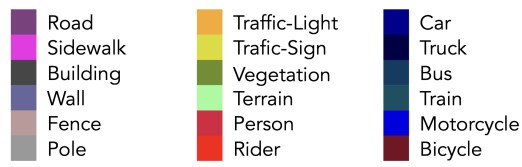

Figure 8: **Color legend for (s4)Cityscapes and (s4)KITTI360** – The driving datasets are colored according to the same color-scheme in this paper to aid comprehension. Note that some classes of Cityscapes do not exist in KITTI360. Also not all classes may be present in the images shown in Fig. 6.

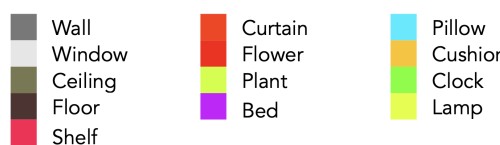

Figure 9: **Color legend for (s4)ADE20K** – Only the classes that are actually present in the segmentation map and ground-truth displayed in Fig. 6 are shown here. Since ADE20K contains 150 classes, providing a full color legend would not be expedient.

## A.2  Parameterization of the Scribble Generation Algorithm

The parameters for the scribble generation algorithm were adjusted according to the resolution and image scales of the respective datasets. For instance, the higher resolution of Cityscapes requires different settings for the $\epsilon$-variables of the algorithm. The exact values are listed in Tab. 3 respectively Tab.,4. The scribble generation code can be found under https://github.com/wbkit/Scribbles4All.

Table 3: **Scribble Generation Parameters for s4Pascal & s4KITTI360** – The parameters are explained in Sec. 3.2. Min. blob size refers to the minimum size of a dense label instance to be assigned a corresponding scribble. $area$ refers to the area of the label instance occupied in pixels from which the scribble is created.

| Parameter | s4Pascal | | s4KITTI360 | |
|---|---|---|---|---|
| $\epsilon_1(x)$ | $\begin{cases} 2 & x \leq 0.003 \\ 2 + \frac{x-0.007}{0.063} & 0.003 \leq x \leq 0.15 \\ 20 & 0.15 \leq x \end{cases}$ | | $\begin{cases} 3 & x \leq 0.007 \\ 3 + \frac{x-0.007}{0.063} & 0.007 \leq x \leq 0.07 \\ 20 & 0.07 \leq x \end{cases}$ | |
| $\epsilon_2$ | 2 | | 3 | |
| $\sigma_{com}$ | $\frac{area}{20}$ | | $\frac{area}{20}$ | |
| $\sigma$ | $\frac{area}{10}$ | | $\frac{area}{10}$ | |
| $N$ | 20 | | 10 | |
| $Q$ | 25 | | 25 | |
| min. blob size | 80 px | | 200 px | |
| line thickness | 3 | | 3 | |

Table 4: **Scribble Generation Parameters for s4Cityscapes & s4ADE20K** – The parameters are explained in Sec. 3.2. Min. blob size refers to the minimum size of a dense label instance to be assigned a corresponding scribble.

| Parameter | s4Cityscapes | | s4ADE20K | |
|---|---|---|---|---|
| $\epsilon_1(x)$ | $\begin{cases} 5 & x \leq 0.007 \\ 5 + \frac{x-0.007}{0.063} & 0.007 \leq x \leq 0.07 \\ 40 & 0.07 \leq x \end{cases}$ | | $\begin{cases} 2 & x \leq 0.003 \\ 2 + \frac{x-0.003}{0.147} & 0.003 \leq x \leq 0.15 \\ 20 & 0.15 \leq x \end{cases}$ | |
| $\epsilon_2$ | 2 | | 2 | |
| $\sigma_{com}$ | $\frac{area}{20}$ | | $\frac{area}{20}$ | |
| $\sigma$ | $\frac{area}{10}$ | | $\frac{area}{10}$ | |
| $N$ | 10 | | 20 | |
| $Q$ | 25 | | 25 | |
| min. blob size | 400 px | | 80 px | |
| line thickness | 5 | | 3 | |

## A.3 Runtime of the Scribble Generation Algorithm

In the following, we list the processing speed of the proposed scribble generation algorithm. The algorithm's image throughput is dependent on the complexity of the processed datasets and the chosen parameters. For the four datasets presented in our work, the processing speed is listed in Tab. 5. Evaluations were performed on a workstation with an Intel Xeon Gold 6144 processor (8 cores w. hyperthreading). The algorithm is parallelizable and performance scales linearly per core. On a single workstation, the conversion of Pascal and Cityscapes is taking less than an hour while the bigger datasets are in the range of a couple of hours. In practice, the algorithm's runtime is secondary as a dataset only needs to be converted once.

Table 5: **Processing speed for the presented datasets** – Average processing speed to process the listed datasets with the scribble generation algorithm to create the s4-Version of those datasets.

| Dataset | Processing Speed |
|---|---|
| PascalVOC | 12.97 img/s |
| ADE20K | 7.43 img/s |
| KITTI360 | 4.53 img/s |
| Cityscapes | 1.23 img/s |

## A.4 Further Dataset Information

This section provides class-wise dataset statistics for the ScribbleSup and s4Pascal datasets for further statistical comparison as listed in Tab. 6. Furthermore, the class-wise label distributions of the s4KITTI360 and s4Cityscapes datasets as visualised in Fig. 4 and Fig. 5 are listed in Tab. 7 respectively Tab. 8.

Table 6: **Class-wise dataset statistics of s4Pascal and ScribbleSup** – The classes are referred to by their index number in the ScribbleSup dataset.

| | | mean | aeroplane 00 | bicycle 01 | bird 02 | boat 03 | bottle 04 | bus 05 | car 06 | cat 07 | chair 08 | cow 09 | diningtable 10 | dog 11 | horse 12 | motorbike 13 | person 14 | pottedplant 15 | sheep 16 | sofa 17 | train 18 | tvmonitor 19 | background 20 |
|---|---|---|---|---|---|---|---|---|---|---|---|---|---|---|---|---|---|---|---|---|---|---|---|
| **ScribbleSup** | Share labelled px. | 2.3 % | 2.1 % | 3.7 % | 3.6 % | 3.5 % | 3.4 % | 3.5 % | 2.2 % | 2.5 % | 2.1 % | 4.2 % | 2.9 % | 2.7 % | 2.3 % | 3.2 % | 2.5 % | 2.9 % | 3.0 % | 3.0 % | 2.7 % | 2.3 % | 2.5 % |
| | 10 px boundary | 11.3 % | 9.2 % | 18.0 % | 22.4 % | 18.3 % | 22.4 % | 23.7 % | 6.8 % | 12.5 % | 4.5 % | 28.0 % | 16.8 % | 15.8 % | 7.6 % | 17.8 % | 13.6 % | 16.1 % | 22.9 % | 17.9 % | 10.9 % | 7.3 % | 10.8 % |
| **s4Pascal** | Share labelled px. | 2.0 % | 1.2 % | 4.3 % | 3.0 % | 3.5 % | 3.7 % | 3.9 % | 2.1 % | 2.6 % | 2.2 % | 3.9 % | 2.7 % | 2.8 % | 2.4 % | 3.0 % | 2.6 % | 3.1 % | 3.0 % | 2.9 % | 2.9 % | 2.5 % | 2.7 % |
| | 10 px boundary | 10.1 % | 8.4 % | 19.7 % | 25.0 % | 14.3 % | 20.1 % | 10.1 % | 7.6 % | 13.8 % | 5.9 % | 19.7 % | 14.5 % | 14.0 % | 8.4 % | 14.0 % | 13.2 % | 9.6 % | 17.6 % | 16.2 % | 11.2 % | 9.2 % | 13.1 % |

Table 7: **Class-wise dataset statistics of s4KITTI360** – The classes are referred to by their name and index number in the KITTI360 dataset.

| | mean | road 00 | sidewalk 01 | building 02 | wall 03 | fence 04 | pole 05 | traffic-light 06 | traffic-sign 07 | vegetation 08 | terrain 09 | person 10 | car 11 | truck 12 | motorcycle 13 | bicycle 14 | sky 15 |
|---|---|---|---|---|---|---|---|---|---|---|---|---|---|---|---|---|---|
| Share of Dense | 3.1 % | 2.8 % | 5.7 % | 2.5 % | 4.3 % | 3.9 % | 8.6 % | 8.5 % | 7.3 % | 2.1 % | 4.9 % | 6.5 % | 3.3 % | 2.1 % | 4.8 % | 7.8 % | 4.4 % |
| 10px distance | 4.8 % | 0.52 % | 7.31 % | 4.28 % | 8.89 % | 7.73 % | 5.37 % | 0.02 % | 2.14 % | 2.40 % | 11.6 % | 0.52 % | 7.40 % | 0.66 % | 0.22 % | 0.25 % | 3.49 % |

Table 8: **Class-wise dataset statistics of s4Cityscapes** – The classes are referred to by their name and index number in the Cityscapes dataset.

| | mean | road 00 | sidewalk 01 | building 02 | wall 03 | fence 04 | pole 05 | traffic-light 06 | traffic-sign 07 | vegetation 08 | terrain 09 | sky 10 | person 11 | rider 12 | car 13 | truck 14 | bus 15 | train 16 | motorcycle 17 | bicycle 18 |
|---|---|---|---|---|---|---|---|---|---|---|---|---|---|---|---|---|---|---|---|---|
| Share of Dense | 2.26 % | 1.50 % | 5.47 % | 3.30 % | 3.11 % | 4.02 % | 10.81 % | 4.42 % | 9.93 % | 3.47 % | 4.76 % | 6.12 % | 8.0 % | 3.0 % | 5.23 % | 1.01 % | 0.71 % | 0.36 % | 1.59 % | 5.10 % |
| 10px distance | 13.81 % | 1.53 % | 15.00 % | 10.43 % | 9.15 % | 10.70 % | 49.96 % | 18.60 % | 33.74 % | 8.55 % | 14.92 % | 6.67 % | 25.83 % | 9.49 % | 11.10 % | 2.39 % | 1.61 % | 0.89 % | 4.15 % | 15.55 % |

Table 9: **Hyperparameters used for Benchmarking** – The hyperparameters for Pascal were kept as documented by the authors of the reference methods and are therefore not listed below. The optimiser for all methods except the EMA was SGD with momentum. The former was trained with AdamW. *SW* refers to sliding window inference, while *FI* denotes full image inference.

| Dataset | Method | lr | BS | train_crop | inference |
|---------|--------|-----|-----|------------|-----------|
| *s4KITTI360* | EMA | 6e-5 | 8 | (376,512) | SW |
| | SASformer | 1e-4 | 8 | (512,512) | SW |
| | TEL | 0.005 | 8 | (376,512) | SW |
| | AGMM-SASS | 0.003 | 8 | (512,512) | FI |
| *s4Cityscapes* | EMA | 4e-5 | 2 | (512, 1024) | SW |
| | SASformer | 0.001 | 4 | (512,512) | SW |
| | TEL | 0.006 | 4 | (512, 1024) | SW |
| | AGMM-SASS | 0.002 | 4 | (721, 721) | FI |
| *s4ADE20K* | EMA | 6e-5 | 8 | (512,512) | FI |
| | SASfomer | 5e-4 | 8 | (512,512) | FI |
| | TEL | 0.001 | 8 | (512,512) | FI |
| | AGMM-SASS | 0.001 | 8 | (512,512) | FI |

## A.5 Further Benchmarking Information

This section documents the hyperparameters used to train the models using the codebases provided by the authors of the reference methods [21, 45, 31]. The main parameters are already provided in Sec. 5.1. This section provides a detailed listing. further details are to be found in Tab. 9.

## A.6 Further Evaluation of EMA method on s4Pascal and ScribbleSup

Additionally, the training of the EMA methods allows for insights into the training process for s4Pascal. As illustrated in Fig. 10 the model's teacher maintains a relative constant certainty with respect to object classes and iteratively refines its prediction certainty on the background class. As the background is present in each image, the model can learn to identify this class and since most Pascal images are one object and the background therefore refine the overall segmentation map. Hence, we conclude that the overall setup of this dataset does not facilitate learning to apprehend inter-object class boundaries as necessary for more complex scenes but rather focuses on the background class. This observation is not exhaustive but provides a possible explanation for the reduced performance gap of scribble-supervised segmentation methods on ScribbleSup and s4Pascal.

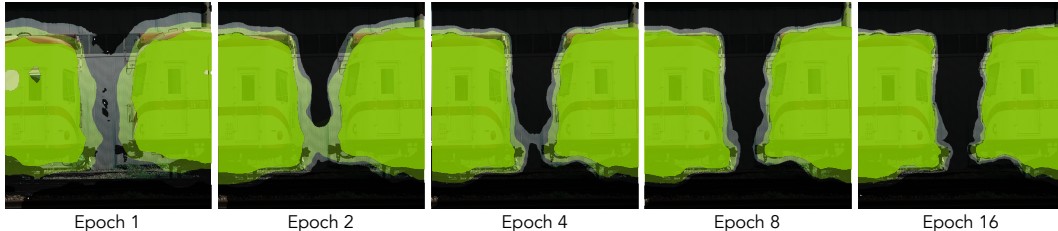

| Epoch 1 | Epoch 2 | Epoch 4 | Epoch 8 | Epoch 16 |

Figure 10: **EMA boundary learning** – The EMA for s4Pascal mainly refines its class boundaries by becoming more confident on the background class. Solid colors are areas where the model is more than 99% certain. Light areas cover the actual prediction where certainty is below that threshold.

# B Dataset Datasheet

## B.1 Motivation

- **For what purpose was the dataset created?** Was there a specific task in mind? Was there a specific gap that needed to be filled? Please provide a description.
  The datasets were created to provide datasets beyond ScribbleSup for the task of scribble-supervised semantic segmentation which is a weakly-supervised segmentation task. It aims to provide more diverse and challenging datasets to researchers in this field adding s4KITTI360 and s4Cityscapes for segmentation in the autonomous driving domain, asADE20K for many-class segmentation and s4Pascal as another scribble set for PascalVOC to verfify and validate our scribble generation algorithm. The datasets facilitates future research into scribble-supervised methods.

- **Who created the dataset (e.g., which team, research group) and on behalf of which entity (e.g., company, institution, organization)?**
  The datasets were created by Wolfgang Boettcher as part of his Master's thesis at ETH Zurich, Switzerland and his PhD research for the Max-Plack Institute for Informatics, Germany.

- **Who funded the creation of the dataset?** If there is an associated grant, please provide the name of the grantor and the grant name and number.
  N/A

- **Any other comments?**
  None.

## B.2 Composition

- **What do the instances that comprise the dataset represent (e.g., documents, photos, people, countries)?** Are there multiple types of instances (e.g., movies, users, and ratings; people and interactions be- tween them; nodes and edges)? Please provide a description.
  The datasets created contain class-wise semantic labels for the aforementioned segmentation datasets. The labels are available as semantic images and coordinate sequences. For the instances covered by the datasets our labels are designed for, refer to the respective documentation of the base datasets.

- **How many instances are there in total (of each type, if appropriate)?**
  The only instances provided are the scribble labels. The statistics of those are documented in the main paper and the first part of the supplementary material.

- **Does the dataset contain all possible instances or is it a sample (not necessarily random) of instances from a larger set?** If the dataset is a sample, then what is the larger set? Is the sample representative of the larger set (e.g., geographic coverage)? If so, please describe how this representativeness was validated/verified. If it is not representative of the larger set, please describe why not (e.g., to cover a more diverse range of instances, because instances were withheld or unavailable).
  The instance coverage is defined by the underlying base datasets. For more details, refer to the respective documentation of the base datasets.

- **What data does each instance consist of?** "Raw" data (e.g., unprocessed text or images) or features? In either case, please provide a description.
  The datasets provided by us exclusively contain labels as semantic images and coordinate sets. For more details, refer to the respective documentation of the base datasets.

- **Is there a label or target associated with each instance?** If so, please provide a description.
  All instances labelled in the underlying base datasets are also labelled in the datasets we created as we derive the scribble labels from the full semantic labels of the base dataset.

- **Is any information missing from individual instances?** If so, please provide a description, explaining why this information is missing (e.g., because it was unavailable). This does not include intentionally removed information but might include, e.g., redacted text.
  The datasets provided by us only contain the labels. For more details, refer to the respective documentation of the base dataset.

- **Are relationships between individual instances made explicit (e.g., users' movie ratings, social network links)?** If so, please describe how these relationships are made explicit.
  The datasets provided by us only contain the labels. For more details, refer to the respective documentation of the base datasets.

- **Are there recommended data splits (e.g., training, development/validation, testing)?** If so, please provide a description of these splits, explaining the rationale behind them.
  We strongly recommend using the same data splits as the base datasets we created the scribble datasets for. For more details, refer to the respective documentation of the base datasets.

- **Are there any errors, sources of noise, or redundancies in the dataset?** If so, please provide a description.
  The scribble labels are created from the semantic masks of the base datasets. Therefore, noise or labelling errors of the datasets may have propagated to our scribble labels. Since most label noise occurs at class edges and our scribble generation algorithm erodes the edges before creating scribbles, it stands to assume that the label noise in our datasets is smaller or equal to the base dataset. We have sanity checked the created scribble labels such that we can conclude that our algorithm does not introduce additional labelling errors and is consistent with the original semantic labels.

- **Is the dataset self-contained, or does it link to or otherwise rely on external resources (e.g., websites, tweets, other datasets)?** If it links to or relies on external resources, a) are there guarantees that they will exist, and remain constant, over time; b) are there official archival versions of the complete dataset (i.e., including the external resources as they existed at the time the dataset was created); c) are there any restrictions (e.g., licenses, fees) associated with any of the external resources that might apply to a dataset consumer? Please provide descriptions of all external resources and any restrictions associated with them, as well as links or other access points, as appropriate.
  The datasets we provide only contain the scribble-labels for already existing segmentation datasets. Therefore, the dataset is not self-contained and relies on external sources for the input data and fully annotated validation/test data. The external datasets our scribble datasets rely on are all popular and established datasets in the semantic segmentation community that have been actively maintained in the past (Cityscapes, KITTI360, ADE20K). Therefore, the persistence of these datasets is save to assume although there are no formal guarantees.

  Due to the varying licenses associated with the different base datasets, we do not provide a "complete" dataset with our labels and the external content. Users will have to download the base dataset first and then integrate the scribble datasets into the respective dataset structures. Documentation on how to do this will be provided on the project GitHub page.

  All external base datasets are subject to the respective license they have been published under. Users will have to check their eligibility with respect to the base datasets on their own and adhere to their terms. The links to the license terms of the four base datasets are listed below. ADE20K was published under the CC-BSD3 license. KITTI360 was publsihed under CC BY-NC-SA 3.0. Cityscapes was published under a proprietary license which limits use to academia and requires citation of the Cityscapes paper. Therefore, we recommend checking that license manually if in doubt.
  http://host.robots.ox.ac.uk/pascal/VOC/
  https://www.cvlibs.net/datasets/kitti-360/
  https://www.cityscapes-dataset.com/license/
  https://groups.csail.mit.edu/vision/datasets/ADE20K/terms/

- **Does the dataset contain data that might be considered confidential (e.g., data that is protected by legal privilege or by doctor-patient confidentiality, data that includes the content of individuals' non-public communications)?** If so, please provide a description.
  The scribble labels provided by us contain none of the aforementioned types of restricted data.

- **Does the dataset contain data that, if viewed directly, might be offensive, insulting, threatening, or might otherwise cause anxiety?** If so, please describe why. The scribble labels provided by us contain no offensive, insulting, threatening content or data that could cause anxiety.

### B.3 Collection Process

- **How was the data associated with each instance acquired?** Was the data directly observable (e.g., raw text, movie ratings), reported by subjects (e.g., survey responses), or indirectly inferred/derived from other data (e.g., part-of-speech tags, model-based guesses for age or language)? If the data was reported by subjects or indirectly inferred/derived from other data, was the data validated/verified? If so, please describe how.
  The datasets provided by us only contain the labels, which are created using the procedure described in the paper. For more details on the input data, refer to the respective documentation of the base datasets.

- **What mechanisms or procedures were used to collect the data (e.g., hardware apparatuses or sensors, manual human curation, software programs, software APIs)?** How were these mechanisms or procedures validated?
  The scribble labels in our datasets were created from the dense segmentation labels of the underlying base datasets by the algorithm presented in the main paper. The main paper also thoroughly validates this generation algorithm by the statistical and functional properties of the created datasets.

- **If the dataset is a sample from a larger set, what was the sampling strategy (e.g., deterministic, probabilistic with specific sampling probabilities)?**
  Our dataset contains scribble labels for the entirety of the labelled set of the underlying base datasets.

- **Who was involved in the data collection process (e.g., students, crowdworkers, contractors) and how were they compensated (e.g., how much were crowdworkers paid)?**
  The datasets were created by the main author himself as part of his master's thesis and regular PhD work in accordance with the compensation standards of ETH Zurich respectively Max-Planck Institute for Informatics.

- **Over what timeframe was the data collected?** Does this timeframe match the creation timeframe of the data associated with the instances (e.g., recent crawl of old news articles)? If not, please describe the time- frame in which the data associated with the instances was created.
  The datasets provided by us only contain the labels. For more details on the timeframe of input data capture, refer to the respective documentation of the base datasets.

- **Were any ethical review processes conducted (e.g., by an institutional review board)?** If so, please provide a description of these review processes, including the outcomes, as well as a link or other access point to any supporting documentation.
  Our research was not subjected to an ethical review process due to the nature of our work and the respective guidelines at ETH Zurich and MPI for Informatics.

### B.4 Preprocessing/cleaning/labeling

- **Was any preprocessing/cleaning/labeling of the data done (e.g., discretization or bucketing, tokenization, part-of-speech tagging, SIFT feature extraction, removal of instances, processing of missing values)?** If so, please provide a description. If not, you may skip the remaining questions in this section.
  The datasets provided by us only contain the labels. For more details on the processing of input data capture, refer to the respective documentation of the base datasets.

- **Was the "raw" data saved in addition to the preprocessed/cleaned/labeled data (e.g., to support unanticipated future uses)?** If so, please provide a link or other access point to the "raw" data.
  N/A

- **Is the software that was used to preprocess/clean/label the data available?** If so, please provide a link or other access point.
  Yes, the algorithm to create our labels is publicly available on Github.

- **Any other comments?**
  None

## B.5   Uses

- **Has the dataset been used for any tasks already?** If so, please provide a description.
  The datasets have been used to benchmark existing state-of-the-art scribble-supervised semantic segmentation methods.

- **Is there a repository that links to any or all papers or systems that use the dataset?** If so, please provide a link or other access point.
  All data and code that relates to the dataset are available from https://github.com/wbkit/Scribbles4All.

- **What (other) tasks could the dataset be used for?**
  The dataset is usable for all training methods for sparsely supervised semantic segmentation. For instance, point label methods can be tested on the dataset as well. Furthermore, the dataset can be used for finetuning vision foundation models to a specific domain.

- **Is there anything about the composition of the dataset or the way it was collected and preprocessed/cleaned/labeled that might impact future uses?** For example, is there anything that a dataset consumer might need to know to avoid uses that could result in unfair treatment of individuals or groups (e.g., stereotyping, quality of service issues) or other risks or harms (e.g., legal risks, financial harms)? If so, please provide a description. Is there anything a dataset consumer could do to mitigate these risks or harms?
  To our knowledge, no aspect of our work has exposure to the aforementioned risk or limitations.

- **Are there tasks for which the dataset should not be used?** If so, please provide a description.
  To our knowledge, there are no specific tasks the dataset should specifically not be used. If in doubt, refer to the documentation of the underlying base datasets.

- **Any other comments?**
  None.

## B.6   Distribution

- **Will the dataset be distributed to third parties outside of the entity (e.g., company, institution, organization) on behalf of which the dataset was created?** If so, please provide a description.
  The dataset is made available to the general public under the stated license to use via GitHub under the CC BY 4.0 license. The links for access can be found here: https://github.com/wbkit/Scribbles4All.

- **How will the dataset will be distributed (e.g., tarball on website, API, GitHub)?** Does the dataset have a digital object identifier (DOI)?
  The central access point for all items related to the datasets is the project's GitHub repository (https://github.com/wbkit/Scribbles4All.) The datasets are currently stored as tarballs on the public GitHub repository and can be downloaded from there.

- **When will the dataset be distributed?**
  The dataset has been made available with the submission to NeurIPS 2024 Datasets and Benchmarks.

- **Will the dataset be distributed under a copyright or other intellectual property (IP) license, and/or under applicable terms of use (ToU)?** If so, please describe this license and/or ToU, and provide a link or other access point to, or otherwise reproduce, any relevant licensing terms or ToU, as well as any fees associated with these restrictions.
  The datasets are distributed under the CC BY 4.0 license. As mentioned above, necessary base datasets may have different license terms. It is up to the user to evaluate if their use case is permissive.

- **Have any third parties imposed IP-based or other restrictions on the data associated with the instances?** If so, please describe these restrictions, and provide a link or other access point to, or otherwise reproduce, any relevant licensing terms, as well as any fees associated with these restrictions. Our dataset provides scribble labels for underlying base datasets. The dataset of the base datasets may be subject to restrictions listed in the respective dataset licenses. Those can be found under the following links:

http://host.robots.ox.ac.uk/pascal/VOC/
https://www.cvlibs.net/datasets/kitti-360/
https://www.cityscapes-dataset.com/license/
https://groups.csail.mit.edu/vision/datasets/ADE20K/terms/

- **Do any export controls or other regulatory restrictions apply to the dataset or to individual instances?** If so, please describe these restrictions, and provide a link or other access point to, or otherwise reproduce, any supporting documentation.
N/A

- **Any other comments?**
None.

## B.7 Maintenance

- **Who will be supporting/hosting/maintaining the dataset?**
The dataset and repository will be maintained by the main author of the paper.

- **How can the owner/curator/manager of the dataset be contacted (e.g., email address)?**
As the owner/maintainer is the main author of the paper, he can be contacted via the e-mail address provided in the main paper (wolfgang.boettcher@mpi-inf.mpg.de). Additionally, the project team can be reached through the issue tracker of the provided GitHub repository.

- **Is there an erratum?** If so, please provide a link or other access point.
Currently, there is no erratum.

- **Will the dataset be updated (e.g., to correct labeling errors, add new instances, delete instances)?** If so, please describe how often, by whom, and how updates will be communicated to dataset consumers (e.g., mailing list, GitHub)?
The dataset may be updated by the maintainer in case of labelling errors or changes in the base datasets our datasets are based on. There will be no regular update interval but updates on request in case of errors. Changes will be announced on the GitHub repository of the datasets as this repository serves as the central access point to all aspects of the project.

- **If the dataset relates to people, are there applicable limits on the retention of the data associated with the instances (e.g., were the individuals in question told that their data would be retained for a fixed period of time and then deleted)?** If so, please describe these limits and explain how they will be enforced.
N/A

- **Will older versions of the dataset continue to be supported/hosted/maintained?** If so, please describe how. If not, please describe how its obsolescence will be communicated to dataset consumers.
The datasets will be incorporated into the GitHub repository of the project alongside the scribble generation algorithms. All changes are transparent through git commits. Changes in the dataset will be announced in the README as well. As dataset changes are primarily planned to remedy labelling errors or similar issues, older versions will not be actively maintained.

- **If others want to extend/augment/build on/contribute to the dataset, is there a mechanism for them to do so?** If so, please provide a description. Will these contributions be validated/verified? If so, please describe how. If not, why not? Is there a process for communicating/distributing these contributions to dataset consumers? If so, please provide a description.
If others want to contribute and modify the dataset, there are no formalised procedures in place since it is not expected to have bigger changes in the lifecycle of the dataset. Contributions are nevertheless welcome and can be requested and coordinated by contacting the main author via e-mail or preferably by using the issue tracking tool of the provided GitHub repository.

- **Any other comments?**
None.

