# OpenReview forum: "Scribbles for All: Benchmarking Scribble Supervised Segmentation Across Datasets"
_NeurIPS.cc/2024/Datasets_and_Benchmarks_Track — NeurIPS 2024 Track Datasets and Benchmarks Spotlight_

### Official Review · Reviewer_ggFR · 2024-06-30
**This paper proposes new scribble datasets and generative algorithms that automatically provide scribble annotations for semantically segmented datasets with pixel-level annotations.**

**Rating:** 7
**Confidence:** 5
**Clarity:** Need improvement

**Review:**

The scribble annotation generation algorithm and the representative scribble dataset proposed in this paper are innovative and of great significance for the development of weakly supervised image semantic segmentation.


The paper is clearly structured, the experiments are well analysed and of good quality. However, it needs to be improved in some places. For example, some redundant phrases such as "with respect to" (line 16) should be avoided. In addition, there are some grammatical errors that need to be improved, such as "made" should be changed to "has made" in line 33, and "show" should be changed to "shown" in line 39.


As the authors point out, the algorithm provided in this paper can only provide scribble tags for datasets that already have pixel-level annotations. Does that mean that if we want to directly obtain scribble annotations for the data, we still need to annotate it at pixel level first? In this case, the cost advantage of scribble annotations is diminished, so does this mean that the algorithm will not be easily applicable to the "application possibilities for custom tasks" mentioned by the authors. Also, I think the authors should evaluate the computational complexity of this algorithm so that it can be subsequently optimised.

**Strengths:**

This work not only provides a scribble annotation dataset, but also proposes an algorithm that can automatically generate scribble labels, and the authors have shown experimentally that the semantic segmentation model trained on synthetic annotations provided in this paper shows excellent competitiveness compared with the model trained on pixel-level annotations, which is very important and innovative for the development of weakly-supervised image segmentation.

In addition, the experiments in this paper are well analysed, and the figures and tables are brief and clear, which can effectively support the ideas or conclusions.

**Additional Feedback:**

See Review

**Correctness:**

The submission and construction are correct. Evaluation methods and experimental design are appropriate and correct.

**Documentation:**

Need improvement

**Ethics:**

No suspect

**Limitations:**

See Review

**Opportunities For Improvement:**

The language in this paper needs to be scrutinized for improvement.

**Relation To Prior Work:**

Yes

**Summary And Contributions:**

This paper not only provides the second scribble dataset to date, but also proposes generative algorithms that can automatically provide scribble annotations for semantic segmentation datasets. This work alleviates the scarcity of segmentation datasets with scribble annotations and lays the foundation for new research in the field of weakly supervised segmentation.

---

> ### Author Rebuttal · Authors · 2024-08-16
>
> We thank the reviewer for his thorough review of our paper and acknowledgement of our contribution to alleviating the scarcity of scribble datasets for semantic segmentation.
>
> **“Does that mean that if we want to directly obtain scribble annotations for the data, we still need to annotate it at pixel level first?”** \
> Our algorithm requires pixel-wise labels in closed shapes to generate the scribbles. In that sense, its applications lie in the conversion of existing and established datasets to the scribble label format as outlined in the general rebuttal.
>
> **“(Does that mean that if we want to directly obtain scribble annotations for the data, we still need to annotate it at pixel level first?) In this case, the cost advantage of scribble annotations is diminished”** \
> We agree that creating dense labels for a new dataset to be then converted into scribbles is not a viable course of action. However, this is not what is intended. Our approach allows us to leverage the vast amount of existing datasets for research into scribble-supervised segmentation methods. Then, methods developed with the help of these datasets can be applied to tasks where scribbles have been obtained via hand annotation. In this setting, scribbles have a clear time advantage over acquiring dense annotations [7, 8].
>
> **Algorithm runtime** \
> The algorithm’s image throughput is dependent on the complexity of the processed datasets and the chosen parameters. For the four datasets presented in our work, the processing speed is as follows:
> | Dataset    | Processing Speed |
> | -------- | -------: |
> | PascalVOC  | 12.97 img/s   |
> | ADE20K | 7.43 img/s    |
> | KITTI360    | 4.53 img/s  |
> | Cityscapes | 1.23 img/s |
>
> on a workstation with an Intel Xeon Gold 6144 processor (8 cores w. hyperthreading). The algorithm is parallelizable and performance scales linearly per core. On a single workstation, the conversion of Pascal and Cityscapes is taking less than an hour while the bigger datasets are in the range of a couple of hours. In practice, the algorithm’s runtime is secondary as a dataset only needs to be converted once.
>
> **Orthography and other remarks** \
> We thank the reviewer for his attention to detail wrt. the orthography of the paper and will pay particular attention to these recommendations, especially regarding redundancy in linking phrases. Furthermore, we will further improve the documentation accompanying the datasets and code.
>
> [7] Di Lin, Jifeng Dai, Jiaya Jia, Kaiming He, and Jian Sun. Scribblesup: Scribble-supervised convolutional networks for semantic segmentation. In Proceedings of the IEEE conference on computer vision and pattern recognition, pages 3159–3167, 2016. Dataset URL: https: //jifengdai.org/downloads/scribble_sup/, Dataset License: http://www.flickr. com/terms.gne?legacy=1. \
> [8] Ozan Unal, Dengxin Dai, and Luc Van Gool. Scribble-supervised lidar semantic segmentation. In Proceedings of the IEEE/CVF Conference on Computer Vision and Pattern Recognition, pages 2697–2707, 2022.

---

> ### Comment · Reviewer_ggFR · 2024-08-28
> **All concerns were responded to positively**
>
> I thank the authors for their detailed response. The rebuttal reasonably addressed my concerns. I trust the authors will update the manuscript appropriately with their rebuttal comments.

---

### Official Review · Reviewer_anV4 · 2024-07-12
**Review for "Scribbles for All: Benchmarking Scribble Supervised Segmentation Across Datasets"**

**Rating:** 6
**Confidence:** 3
**Correctness:** This dataset is constructed soundly.
**Clarity:** The paper is clear and easy to follow.

**Review:**

This paper effectively presents an algorithm for generating scribble labels for semantic segmentation datasets and applies it to several popular segmentation datasets. However, the algorithm is significantly constrained by the necessity of using semantic segmentation ground truth while generating scribble labels, which limits its broader applicability.

**Strengths:**

1. The proposed annotation method can automatically generate scribble labels for the dataset, significantly reducing the workload of manual annotation.
2. The statistical data and analysis for the generated dataset are presented with clear and concise detail.

**Additional Feedback:**

Overall, while this paper introduces automatic annotation of scribble labels for weakly supervised semantic segmentation research, the methodology's limitations are significant, and it lacks substantial generalizability.

**Documentation:**

There are details on data organization, availability, and maintenance.

**Ethics:**

No clear ethical concerns with the submission warrant further discussion or review.

**Limitations:**

The authors should consider applying the generation method of scribble labels to entirely unlabeled datasets.

**Opportunities For Improvement:**

1. As mentioned in the Review part, the proposed annotation generation method requires semantic segmentation ground truth, which limits the scale of weakly supervised data generation. In contrast, the advantage of weakly supervised data lies in its large-scale availability.
2. Several existing methods provide weak supervision for semantic segmentation, such as generating pseudo-labels with the segment anything model [1]. What are the advantages of the proposed method over these existing approaches?
3. Could multiple scribble labels be generated in one segment?

[1] Kirillov A, Mintun E, Ravi N, et al. Segment anything. Proceedings of the IEEE/CVF International Conference on Computer Vision. 2023: 4015-4026.

**Relation To Prior Work:**

Yes.

**Summary And Contributions:**

This article proposes an algorithm for automatically generating scribble labels for semantic segmentation datasets. It also generates scribble labels for several popular segmentation datasets and evaluates and analyzes the corresponding algorithms.

---

> ### Author Rebuttal · Authors · 2024-08-16
>
> We thank the reviewer for his discerning review and acknowledgement of the thoroughness of analysis and statistical data for our datasets. In the following, we will address the main concerns voiced.
>
> **“The proposed annotation generation method requires semantic segmentation ground truth, which limits the scale of weakly supervised data generation. In contrast, the advantage of weakly supervised data lies in its large-scale availability.”** \
> We do agree with the reviewer that the size of the underlying supervised segmentation datasets limits the size of the scribble datasets that can be produced through our pipeline. Having said this, this procedure entails the advantage of having direct comparability with the respective supervised datasets and the methods benchmarked on it.
> Moreover, common practice moves towards large-scale pretraining and general-purpose models that are subsequently fine-tuned on relatively small datasets. Our work provides the datasets to advance research on how to do that, using low-cost scribble annotations. Creating large-scale semi-supervised datasets is a valuable endeavor but outside the scope and intention of our paper.
>
> **“What are the advantages of the proposed method over these existing approaches?”(ex. SAM)** \
> Generating pseudo labels for weakly supervised segmentation has become more popular recently as noted by the reviewer. In our view, these methods do not directly compete with providing scribble labeled datasets due to the following reasons:
> 1. SAM is a mask generator that was trained in a supervised fashion on a dataset containing about 1 billion masks. Using SAM is not a strictly weakly supervised setup.
> 2. Incorporating SAM for pseudo-label generation requires prompting. While this can be done with other means of weak supervision such as points, investigations indicate that scribbles lead to better SAM pseudo-labels [1]. Pseudo Label generators and scribble-labeled datasets are therefore complementary.
> 3. SAM is not universally applicable to any dataset since the generated masks deteriorate when domain gaps to SAM’s training data are present. For instance, SAM is known to fail in certain medical imaging scenarios [2, 3]. For such cases, learning methods that directly utilize scribble labels of the domain-specific data are still required and also necessary to enable future research in that direction.
>
> **“The authors should consider applying the generation method of scribble labels to entirely unlabeled datasets.”** \
> Our method is not intended to be used on unlabeled datasets as discussed in the general rebuttal. Generating scribbles for semantic segmentation without any annotations would constitute unsupervised semantic segmentation, which is another direction of research.  The general setup there is to do pretraining on large-scale unsupervised datasets to learn suitable spatial features. Yet, those methods still employ a second step of lightweight supervision through labels, clustering or  prompting and still fail to achieve similar performance to weakly-supervised methods [4,5,6]. The current SOTA is still about 10 % mIoU worse on PascalVOC than the SOTA scribble-supervised method without pretraining.
>
> **“Could multiple scribble labels be generated in one segment?”** \
> As described in the Methodology section of the paper, the algorithm separates delicate segments into multiple blobs that each get a scribble label assigned. Therefore, multiple scribble labels may be generated per segment. If explicitly desired, the parameters of the scribble generation algorithm may be altered such that multiple labels per object could be generated.
>
> [1] Peng-Tao Jiang and Yuqi Yang, Segment Anything is A Good Pseudo-label Generator
> for Weakly Supervised Semantic Segmentation, https://arxiv.org/pdf/2305.01275 \
> [2] Yuqing Wang, Yun Zhao, Linda Petzold, An empirical study on the robustness of the segment anything model (SAM), Pattern Recognition, Volume 155, 2024, 110685, ISSN 0031-3203, https://doi.org/10.1016/j.patcog.2024.110685. \
> [3] Maciej A. Mazurowski, Haoyu Dong, Hanxue Gu, Jichen Yang, Nicholas Konz, Yixin Zhang, Segment anything model for medical image analysis: An experimental study, Medical Image Analysis, Volume 89, 2023, 102918, ISSN 1361-8415, https://doi.org/10.1016/j.media.2023.102918. \
> [4] Y. Liu, J. Zeng, X. Tao and G. Fang, "Rethinking Self-Supervised Semantic Segmentation: Achieving End-to-End Segmentation," in IEEE Transactions on Pattern Analysis and Machine Intelligence, doi: 10.1109/TPAMI.2024.3432326. \
> [5] K. Li et al., "ACSeg: Adaptive Conceptualization for Unsupervised Semantic Segmentation," 2023 IEEE/CVF Conference on Computer Vision and Pattern Recognition (CVPR), Vancouver, BC, Canada, 2023, pp. 7162-7172, doi: 10.1109/CVPR52729.2023.00692. \
> [6] M. Caron, N. Houlsby and C. Schmid, "Location-Aware Self-Supervised Transformers for Semantic Segmentation," 2024 IEEE/CVF Winter Conference on Applications of Computer Vision (WACV), Waikoloa, HI, USA, 2024, pp. 116-126, doi: 10.1109/WACV57701.2024.00019.

---

> > ### Comment · Reviewer_anV4 · 2024-08-29
> >
> > I have updated the score.

---

### Official Review · Reviewer_caBU · 2024-07-25

**Rating:** 9
**Confidence:** 4
**Correctness:** Yes
**Clarity:** Yes

**Review:**

This paper presents an automatic scribble generator for any fully labeled segmentation dataset, especially four popular semantic segmentation datasets officially. An an automatic scribble generator is proposed for any fully labeled segmentation dataset, so the contribution does not limited to the four proposed datasets. Moreover, the four datasets serve as strong benchmark to evaluate methods for scribble supervised semantic segmentation. Experiments are solid to show the similarity of generated scribbles to human scribbles.
However, this is only an algorithm for dataset converting, but not scribble generation from scratch or from other weak annotation, so technical contribution may be limited.

**Strengths:**

An an automatic scribble generator is proposed for any fully labeled segmentation dataset, so the contribution does not limited to the four proposed datasets. Moreover, the four datasets serve as strong benchmark to evaluate methods for scribble supervised semantic segmentation.
Experiments are solid to show the similarity of generated scribbles to human scribbles.

**Additional Feedback:**

No additional feedback

**Documentation:**

Yes

**Ethics:**

No ethical concerns

**Limitations:**

Yes, limitations are addressed in the draft

**Opportunities For Improvement:**

This is only an algorithm for dataset converting, but not scribble generation from scratch or from other weak annotation, so technical contribution may be limited.

**Relation To Prior Work:**

Relation to prior work is properly addressed

**Summary And Contributions:**

This paper presents an automatic scribble generator for any fully labeled segmentation dataset, especially four popular semantic segmentation datasets officially.

---

> ### Author Rebuttal · Authors · 2024-08-16
>
> We thank the reviewer for his insightful review and affirmation of the general usability of our algorithm with pre-existing segmentation datasets.
>
>
> **However, this is only an algorithm for dataset converting, but not scribble generation from scratch** \
> The topic of scribble generation from scratch is discussed by us in the general rebuttal.
>
>
> **Scribble generation [...] from other weak annotation** \
> It is correct that our method requires weak annotation that covers a subset of the object with no overlap with other objects (dense, coarse or inside bounding box). For instance, it could also generate scribble labels from the coarse segmentation annotations of Cityscapes. Given the large amount of existing pixel-wise annotated segmentation datasets, our method opens up research into scribble-supervised methods on all these datasets.

---

### Author Rebuttal · Authors · 2024-08-16

We thank the reviewers for their very positive feedback and suggestions. We appreciate that our work was found to be “innovative and of great significance for the development of weakly supervised image semantic segmentation” (ggFR) through “alleviating the scarcity of segmentation datasets with scribble annotation” (ggFR). Further, it is confirmed that our label generation algorithm proposed to achieve the aforementioned advancements generalizes beyond the datasets provided with this paper (caBU, ggFR) and that the contributed datasets are a “strong benchmark to evaluate methods for scribble supervised semantic segmentation” (caBU). We are also appreciative that our experiments and evaluations are seen by all reviewers (caBU, anV4, ggFR) as clearly structured and well-analyzed. In the following, the common questions of the reviewers are addressed below and specific inquiries are covered in the individual rebuttals. We highly appreciate the provided feedback and will incorporate it into the final paper version.

**(caBU, anV4, ggFR) Necessity of Segmentation Ground Truth for Scribble Generation** \
Scribble labels provide a powerful yet low-annotation cost way to provide weak supervision for segmentation datasets. However, prior to our work, there was only one dataset with scribble annotations available, limiting a reliable benchmarking process of scribble-supervised methods.
The primary focus of our work is to automatically derive several scribble-supervised datasets from established segmentation datasets. Hereby, we enable the research community to train and benchmark methods for scribble weakly supervised semantic segmentation (WSSS) thoroughly and fairly against fully supervised methods on the same data. Hence, the scribble generation algorithm is designed to use dense ground truth as the input for scribble generation. It alleviates the necessity to create hand-generated labels for existing segmentation datasets and is applicable to the entirety of supervised segmentation datasets.
Our work does NOT aim to replace manual labeling for new unlabelled datasets. Here, the user will still need to provide manual scribble annotations. However, these have a much lower cost than traditional full semantic segmentation annotations.

---

### Decision · Program_Chairs · 2024-09-26

**Decision:**

Accept (Spotlight)

**Comment:**

All reviewers have expressed their willingness to accept this paper, recognizing its contributions and significance.  I recommend that the authors make necessary revisions in the final version according to the suggestions, and tend to accept this work.